# Robust EEG Based Biomarkers to Detect Alzheimer’s Disease

**DOI:** 10.3390/brainsci11081026

**Published:** 2021-07-31

**Authors:** Ali H. Al-Nuaimi, Marina Blūma, Shaymaa S. Al-Juboori, Chima S. Eke, Emmanuel Jammeh, Lingfen Sun, Emmanuel Ifeachor

**Affiliations:** 1School of Engineering, Computing and Mathematics, Faculty of Science and Engineering, University of Plymouth, Drake Circus, Plymouth PL4 8AA, UK; shaymaa.al-juboori@plymouth.ac.uk (S.S.A.-J.); chimastanley.eke@plymouth.ac.uk (C.S.E.); emmanuel.jammeh@plymouth.ac.uk (E.J.); L.Sun@plymouth.ac.uk (L.S.); E.Ifeachor@plymouth.ac.uk (E.I.); 2College of Education for Pure Science (Ibn Al-Haitham), University of Baghdad, Al Adhamiya, Baghdad 10053, Iraq; 3Department of Physiology and Pharmacology “Vittorio Erspamer”, Sapienza University of Rome, Piazzale Aldo Moro 5, 00185 Rome, Italy; marina.bluma@uniroma1.it

**Keywords:** robust EEG based biomarkers, detection of Alzheimer’s disease, slowing of the EEG, reduction in EEG connectivity, reduction in EEG complexity

## Abstract

Biomarkers to detect Alzheimer’s disease (AD) would enable patients to gain access to appropriate services and may facilitate the development of new therapies. Given the large numbers of people affected by AD, there is a need for a low-cost, easy to use method to detect AD patients. Potentially, the electroencephalogram (EEG) can play a valuable role in this, but at present no single EEG biomarker is robust enough for use in practice. This study aims to provide a methodological framework for the development of robust EEG biomarkers to detect AD with a clinically acceptable performance by exploiting the combined strengths of key biomarkers. A large number of existing and novel EEG biomarkers associated with slowing of EEG, reduction in EEG complexity and decrease in EEG connectivity were investigated. Support vector machine and linear discriminate analysis methods were used to find the best combination of the EEG biomarkers to detect AD with significant performance. A total of 325,567 EEG biomarkers were investigated, and a panel of six biomarkers was identified and used to create a diagnostic model with high performance (≥85% for sensitivity and 100% for specificity).

## 1. Introduction

Alzheimer’s disease is a neurodegenerative, progressive, irreversible, and fatal disease of the brain cells [1,2]. It is characterised by gradual cognitive impairments accompanied by abnormal behaviour, loss of memory, and personality changes [1,3,4]. The two main neuropathologic hallmarks of AD are extracellular amyloid beta (Aβ) plaques and intracellular neurofibrillary tangles (NFTs). The production of Aβ, which represents a vital stage in AD pathogenesis, is the result of an aberrant cleavage of the amyloid peptide precursor protein (APP) which is overexpressed in AD [5,6,7]. In histopathological terms, AD is characterised by the accumulation of senile plaques and neurofibrillary tangles [8]. The senile plaques consist mainly of β-amyloid peptides, while the fibrillary tangles consist of abnormal hyperphosphorylated insoluble forms of the TAU-protein [9,10]. AD is the most common form of dementia (others include vascular, Lewy body, or frontotemporal dementia) and accounts for between 60% to 80% of all dementias worldwide [2,11,12]. Age is the main risk factor for AD [9]. Loss of recent memory is one of the first symptoms of AD (early stage), followed by mild cognitive impairment (MCI), and then severe AD, which is the advanced stage [1]. MCI describes a transitional cognitive state between normal ageing and dementia [9,13] and has been proposed as a disease describing elderly people with mild cognitive impairment but not dementia [13,14]. However, only 80% of MCI cases go on to develop dementia [9,15], although this is sufficient to identify people at high risk of dementia [13]. At the macroscopic level, severe atrophy of the brain is the most common characteristic of AD patients, leading to enlargement of the ventricular system and shrinkage of cortical sulci [16]. In the preclinical stages of AD, the atrophy primarily affects medial temporal parts of the brain, including hippocampal formation [16]. Hippocampal atrophy is associated with more severe memory impairment in AD [17]. Such atrophy could therefore be used as a marker of disease development in clinical trials [11]. 

AD is the main cause of disability among older people [18]. At present, there are over 46.8 million individuals with dementia worldwide with an annual cost of care estimated at US $818 billion [19,20]. This is projected to reach 74.7 million by 2030 with an annual cost of US $2 trillion [21] due to the ageing population and represents a major challenge to health and social care services worldwide [22,23]. It is widely accepted that AD diagnosis in its early stages makes it possible for patients to gain access to appropriate health care services and would facilitate the development of new therapies [24,25,26,27]. However, up to 50% of those living with AD may not have received a formal diagnosis [28,29].

The neurodegeneration due to AD pathology starts many years before the onset of clinical symptoms [14,23,30,31,32]. A biomarker that can detect the changes due to AD would be useful in diagnosing AD in the early stages. Different types of AD biomarkers exist, such as those based on neuroimaging, cerebrospinal fluid (CSF) and blood [14,33,34]. However, neuroimaging (e.g., using PET) is expensive and only available at specialist centres [35,36] and CSF involves lumbar puncture which is invasive [3,37,38]. Blood-based biomarkers have shown promise, but they are still under development and are not in routine use [36,39].

The detection of AD at the early stages implies screening large numbers of people. Therefore, there is a need for a common, non-invasive and low-cost, easy to use, and robust “gatekeeper” biomarker for preliminary selection of people at risk of AD pathology before referring them to specialized clinics [23,34,40]. Furthermore, biomarkers such as PET, CSF and MRI may not be suitable for routine serial recordings over time for AD diagnosis and monitoring [41] because of cost or invasiveness.

Potentially, EEG can play a valuable role in fulfilling this need (e.g., as a first-line decision-support tool in AD diagnosis [30]). AD affects numerous subcortical pathways in the brain [42] that lead to changes in the information-processing activity of the brain and these changes are thought to be reflected in the information content of the EEG [43,44,45]. Thus, EEG provides valuable information about changes in electrophysiological brain dynamics due to AD [43,46,47,48,49]. EEG is non-invasive, low-cost, common, and has high sensitivity in discriminating between AD patients and normal elderly (Nold) subjects [50,51,52,53]. The potential utility of EEG to detect brain signal changes even in the early stage of the disease has been demonstrated [54]. 

Promising EEG based biomarkers can be divided into three main categories, namely those based on the slowing of the EEG signal, reduction in EEG complexity, and a decrease in functional connectivity of EEG among cortical regions [47,48,54,55,56,57,58,59,60]. The slowing of the EEG signal [30,48,58,61] is one of the consistent features in AD and this can be quantified as biomarkers [30,58,62]. Several methods are being developed to quantify the slowing of EEG, such as the analysis of changes in the EEG amplitude (ΔEEG_A_), zero-crossing intervals (ZCI) [30], and changes in the power spectrum (ΔPS) of EEG signals [30,47,56,58,59,63,64,65,66,67,68,69,70]. 

EEG coherence is used to measure functional connectivity in the brain [71,72]. AD causes changes in the cortical activity of the brain [73] which impacts the connectivity among cortical regions of the brain [58] and this is reflected in the EEG coherence. [74]. EEG coherence is a sensitive method for assessing the integrity of structural connections between brain areas in AD patients [74]. 

Complexity is a measure of the extent to which the dynamic behavior of a given sequence resembles a random one [75] and is emerging as an important way to detect AD. The cortical areas of the brain fire spontaneously and this dynamic behavior of the brain is complex [76,77]. AD causes a reduction in neuronal activity of the brain [78] resulting in a decrease in the ability of the brain to process information [79,80,81] which may be reflected in the EEG signals [78]. EEG complexity is potentially a promising biomarker [58] as AD patients have a significantly reduced EEG complexity [55,57,58,59,78,82,83]. The reduction in EEG complexity due to AD can be measured using many emerging approaches such as Tsallis entropy (TsEn) [48,61,84], Higuchi Fractal Dimension (HFD) [85,86,87], Approximation Entropy (ApEn) [88,89,90,91], and Lempel Ziv Complexity (LZC). The TsEn, LZC and HFD-based biomarkers are of particular interest. TsEn is one of the most promising information theoretic methods for quantifying EEG complexity [48,61,84,92,93]. Its fast computation may serve as the basis for real-time decision support tools for diagnosing AD [48,84,92,94,95]. HFD is a fast nonlinear computational method for obtaining the fractal dimension of time series signals [96,97,98] and yields biomarkers which are significantly lower in AD patients than in normal subjects [85,86,87]. LZC is a nonparametric and nonlinear method that provides a way to quantify the complexity of the EEG [99,100] and has been used to analyse the EEG complexity in AD [101,102,103]. 

Although there are a large number of existing and new EEG-based biomarkers for AD, no single biomarker is robust enough for use in clinical practice or provides a clear-cut detection of AD in the early stages [3,36,104,105,106]. Robust biomarkers of AD should be consistent and have high detection performance (e.g., in terms of sensitivity and specificity) [3,4,107]. Potentially, a combination of biomarkers could provide the required level of robustness necessary for clinical use [3,23,34,40,108,109]. Few studies have investigated how to combine different EEG biomarkers to exploit their strengths. Hamadicharef et al. [92] developed a logistic regression model to combine AD biomarkers, but the study considered only a small number of specific EEG biomarkers and did not investigate all the possible set of biomarkers. Poil et al. [110] developed a composite EEG based biomarker to predict the conversion of subjects with mild cognitive impairment (MCI) to AD by combining multiple biomarkers. This has a sensitivity of 88% and specificity of 82% and is based mainly on spectral and temporal biomarkers. 

The EEG is a complex signal with a significant amount of information about brain dynamics embedded within it. This, together with a large number of analytical techniques that are used to analyse the EEG and a large number of possible EEG biomarkers has led to uncertainty about how to determine EEG biomarkers to capture AD related processes accurately [111]. To address this, we have developed a methodological framework for the development of robust EEG biomarkers. 

This allows a comprehensive and systematic study of existing and emerging EEG biomarkers associated with AD to reveal the most promising ones, and how to integrate and develop them further to create robust biomarkers. This approach has made it possible to achieve high performance (close to 100% for sensitivity and specificity) and should facilitate acceptance of EEG biomarkers. No such comprehensive EEG biomarker study for AD has hitherto been carried out. The EEG biomarkers which show promising results in AD detection were investigated. A total of 325,567 biomarkers were investigated. Biomarkers associated with EEG features, such as slowing of EEG, reduction in EEG complexity, and a decrease in functional connectivity of EEG among cortical regions, eight analysis methods (i.e., ΔPS, ΔEEG_A_, ZCI, TsEn, HFD, ApEn, LZC, and EEG coherence), and five EEG bands (delta, theta, alpha, beta, and gamma) were selected to construct all the possible combinations of biomarkers. We thought combining multiple biomarkers of different techniques could provide better performance than using each biomarker separately and could reveal hidden separation boundaries. Different studies investigated this hypothesis, but there was no existing study that investigated all possible biomarkers in one study. For example, Poil et al. [110] integrated six EEG biomarkers into a diagnostic index using logistic regression. They concluded that combining different biomarkers could improve the accuracy of predicting conversion from MCI to AD compared with the best individual biomarker in this index. Rossini et al. [109] suggested it could be useful in combination with other markers including EEG connectivity. The proposed study provides a novel framework for constructing robust biomarkers that can be used to detect AD with high performance in terms of sensitivity and specificity by exploiting the combined strengths of different biomarkers. 

The paper is arranged as follows. In Section 2, the methodology is described. The results are presented in Section 3 and discussed in Section 4. The paper is concluded in Section 5.

## 2. Materials and Methods

### 2.1. Materials

The data used in this study was collected from 112 volunteers from previous studies conducted at Derriford Hospital, Plymouth, UK, and at La Sapienza, University of Rome, Italy [30]. The Plymouth data was used to develop the methodological framework and the resulting biomarkers and diagnostic models to detect AD. The independent data from Rome was used to optimize and validate the models. The Plymouth data was collected from 52 volunteers (20 AD patients and 32 Nold subjects) referred to the EEG department in the hospital from a specialist memory clinic. It is the clinical practice at the memory clinic that all patients undergo a battery of psychometric tests (including, MMSE [28], Rey Auditory Verbal Learning Test [29], Benton Visual Retention Test [30], and memory recall tests [31]) before referral. The results from the psychometric tests are scored and interpreted by a specialist psychologist and all clinical and psychometric findings are discussed at a multidisciplinary team meeting following the clinic. Each patient is then referred to the hospital for EEG assessment by the memory clinic with a working diagnosis (e.g., probable AD, depending on the outcome of the clinical and psychometric assessments). All controls were healthy volunteers and had normal EEGs (confirmed by a Consultant Clinical Neurophysiologist). The classification of subjects with dementia is based on the working diagnosis provided by the specialist memory clinic and EEG findings. Magnetic resonance imaging (MRI) data was not recorded because this facility was not available at the hospital at the time. The EEG data was obtained using a strict protocol from Derriford Hospital, Plymouth, U.K., and had been collected using normal hospital practices. The entire collection of recordings includes a variety of states e.g., hyperventilation, awake, drowsy and alert, with periods of eyes closed and open. For storage reasons, the sampling rate was reduced from 256 Hz to 128 Hz by averaging two consecutive samples. The duration of each EEG signal is 4 minutes. Data from a fixed interval (61s to 240s) was used to avoid electrical artefact, which regularly occurs at the beginning of records [30], leaving a standard three-minute data to analyse. The Plymouth data consists of two datasets, datasets A and B. Dataset A includes 11 age-matched subjects over 65 years old (3 AD and 8 healthy controls). It was recorded using the traditional 10–20 system in a common reference montage by using the average of all channels as a reference. The EEG signals were converted to common average and bipolar montages using software. Dataset B includes 41 subjects that were not perfectly age-matched with 24 Nold and 17 were probable AD patients. It was recorded using the modified Maudsley system. The conventional 10–20 system has a similar setting to the Maudsley electrode positioning system [112]. 

In this study, the methodological framework and the resulting biomarkers and models were based on the Plymouth data where neuro-imaging studies were not carried out and for which access to the neuro-psychological scores was not possible because the patients were from a different district, although full cognitive tests were carried out. To optimise and evaluate the method, we have used independent datasets from another centre in Rome (Assessment Datasets C and D), which had available all clinical data including MMSE scores for both AD and healthy controls and the subjects had undergone neuro-imaging studies.

The Rome data consists of 60 volunteers (30 AD patients and 30 Nold subjects). The sampling rate was 128 Hz. The duration of each EEG signal was 1 minute. The EEGs were obtained using the 10–20 system. The Rome data consists of two datasets, dataset C and D. Dataset C includes 20 subjects collected from 10 probable AD patients and 10 age-matched Nold subjects. Dataset D includes 40 subjects collected from 20 age-matched Nold subjects and 20 Probable AD patients. Probable AD was diagnosed according to NINCDS-ADRDA [32] and DSM IV criteria. The patients underwent general medical, neurological and psychiatric assessments. They were also rated with a number of standardized diagnostic and severity instruments that included MMSE [33], Clinical Dementia Rating Scale (CDRS) [34], Geriatric Depression Scale (GDS) [35], Hachinski Ischemic Scale (HIS) [36], and Instrumental Activities of Daily Living scale (IADL) [37]. Neuroimaging diagnostic procedures (computed tomography or MRI) and complete laboratory analyses were carried out to exclude other causes of progressive or reversible dementias, to have a homogenous mild AD patient sample. Exclusion criteria included, in particular, any evidence of (1) frontotemporal dementia, (2) VaD (i.e., VaD was also diagnosed according to NINDS-AIREN criteria; [38], (3) extra-pyramidal syndromes, (4) reversible dementias (including pseudodementia of depression), and (5) fluctuations in cognitive performance (suggestive of a possible Lewy body dementia). The normal control subjects were recruited mainly among patients’ spouses. All Nold subjects underwent physical and neurological examinations as well as cognitive screening (including MMSE). Subjects affected by chronic systemic illnesses (i.e., diabetes mellitus or organ failure) were excluded, as were subjects receiving psychoactive drugs. Subjects with a history of present or previous neurological or psychiatric disease were also excluded. All Nold subjects had a GDS score lower than 14. Table 1 Summarises the EEG datasets that were used in this study.

Figure 1 shows the electrode locations using the 10–20 system. The letters F, P, O, and T refer to cerebral cortex lobes (F: frontal, P: parietal, O: occipital, and T: temporal), C for the central region, Fp (frontopolar), and A refers to ear channel [113]. 

### 2.2. Methodological Framework

Figure 2 provides an overview of the methodological framework for the development of robust EEG biomarkers to detect AD with clinically acceptable performance by exploiting the combined strengths of different EEG based biomarkers.

It supports the investigation, development, integration, and assessment of the performance of new and promising biomarkers based on the three main characteristics of the dementia EEG (slowing of the EEG, reduction in EEG complexity and reduction in EEG coherence). The emphasis is on finding the best possible combination of EEG biomarkers to detect AD accurately. The development of robust and composite EEG biomarkers requires the identification of EEG features which have a significant association with AD. 

The main EEG features are those associated with the slowing of the EEG (e.g., shifts in the EEG power to the lower frequencies), reduction in EEG complexity (reduction in complexity measures) and reduction in EEG coherence. The features were extracted from the five traditional EEG bands (i.e., delta 0–4 Hz, theta 4–8 Hz, alpha 8–12 Hz, beta 12–30 Hz, and gamma 30–45 Hz) and then quantified as biomarkers. Previous studies have shown that biomarkers derived from the EEG frequency bands (instead of the entire EEG) have enhanced performance [55]. The biomarkers are computed based on the slowing of EEG (i.e., ΔPS, ΔEEG_A_, and ZCI), reduction in EEG complexity (i.e., TsEn, HFD, LZC, and ApEn) and reduction in coherence for each of the five EEG frequency bands and for each EEG channel. The biomarkers include those which have shown promise in AD detection with high performance [30,47,48,54,55,56,57,58,59,60,61,62,63,64,65,66,67,68,69,70]. The best performing biomarkers are then found using Support Vector Machine (SVM). SVM was used because of its robustness with high-dimensional data and because it has been shown to perform well in previous AD studies compared to other machine learning methods [114,115,116,117]. A Linear Discriminant Analysis (LDA) was then used to combine the selected biomarkers to create the diagnostic model to detect AD.

The following steps outline the procedure for deriving the EEG biomarkers
Filter the EEG signals into five frequency bands (i.e., delta, theta, alpha, beta, and gamma). For this step, a low computational infinite impulse response (IIR) Chebyshev-II bandpass filters were used for computational efficiency in extracting the biomarkers [118].Compute the biomarkers based on the slowing of EEG (i.e., ΔPS, ΔEEG_A_, and ZCI), reduction in EEG complexity (i.e., TsEn, HFD, LZC, and ApEn) and reduction in EEG coherence.Select EEG biomarkers that have a significant association with AD (in terms of the p-values). The EEG biomarkers with *p*-values ≤ 0.001 between AD and normal were selected.Construct panels of biomarkers from the selected biomarkers to enhance performance.For each panel of biomarkers, develop a machine learning model to detect AD. In this step, we used an SVM method and a 10 fold-cross validation strategy to assess performance.Select EEG biomarker panels with sensitivity and specificity values above a specified threshold (80% in this case).Develop diagnostic models by combining the biomarker panels selected in Step 6 using the LDA classifier.Evaluate the diagnostic models using an unseen dataset. The performance of each model was used as a criterion for the evaluation.Optimise the diagnostic model to use the least possible number of biomarkers and still maintain high performance.Validate the optimised model.

In the framework in Figure 2, two supervised machine learning classification methods were used, the SVM and LDA. LDA is commonly used in classification and provides optimal separation in classification [119,120,121]. It uses the global characteristics of the data instead of local characteristics used in SVM [122]. SVM was used (Step 5) to combine biomarker panels for the same method of biomarker computation e.g., [TsEn(delta), TsEn(theta), TsEn(alpha)] because it uses the local characteristics of the data. LDA was used (Step 7) to combine the EEG biomarkers across different methods (e.g., [LZC(delta), TsEn(theta), HFD(alpha), ΔEEG_A_ (beta), ApEn (gamma)]) [123] because it uses the global characteristics of the data. 

Following the previous approach [16], the complete recordings of the EEG including artefacts were used without a prior selection of data elements for analyses. This enabled us to have an idea about the robustness and usefulness of the methods in practice. Powerful analysis methods in the study such as ΔEEG_A_, ΔPS, ZCI, TsEn, HFD, ApEn, LZC are relatively insensitive to the effects of artefacts. Furthermore, frequency band filtering of the EEG signals also reduces the effects of some artefacts, such as muscle artefacts and eye movements [124,125]. The performance of the biomarkers for AD detection is assessed in terms of sensitivity (Sen), specificity (Spec), accuracy (ACC), F-measure, positive predictive value (PPV), and negative predictive value (NPV). Matthew’s correlation coefficient (MCC) was used to measure the quality of the classification (AD and Nold) between the actual and predicted results [126,127].

#### 2.2.1. Biomarkers Based on EEG Slowing, Reduction in EEG Complexity and Coherence

The EEG signals are first filtered into the five EEG frequency bands using a low computational IIR filter to emphasize the main features. The main EEG features in dementia are those associated with the slowing of the EEG (e.g., shifts in the EEG power to the lower frequencies), reduction in EEG complexity (reduction in complexity measures) and reduction in EEG coherence. These features are quantified as biomarkers in each of the five EEG frequency bands. Biomarkers based on ΔPS, ΔEEG_A_, and ZCI are computed for EEG slowing; for the reduction in EEG complexity, the computed biomarkers are based on TsEn, HFD, LZC, and ApEn; for connectivity, they are based on the coherence values between the channels.

*Changes in the power spectrum (ΔPS):* ΔPS [128,129] biomarker computation is based on the magnitude square of the Fast Fourier transform (FFT) of an N-sample EEG data sequence *x*(1), *x*(2), …, *x*(*N*). The power spectrum shows the coefficients for each frequency measured by the FFT.
(1)$$\Delta {\mathrm{PS}}_{X\left(N\right)}={\left[\left|\mathrm{FFT}\left(X\left(N\right)\right)\right|\right]}^{2}$$

*Changes in the EEG amplitude (ΔEEG_A_):* ΔEEG_A_ [23] is used as a measure of the EEG slowing. ΔEEG_A_ is the sum of the differences between adjacent amplitudes of EEG values over the duration of the signal in a second. ΔEEG_A_ of an N-sample data sequence *x*(1), *x*(2), …, *x*(*N*) is calculated as;
Partition the signal X into K partitions, where *K* = *N*/*Fs*.Calculate the ΔEEG_A_ for each partition as; (2)$$\Delta {\mathrm{EEG}}_{A_{k}}=\frac{\sum^{\text{​}}\Delta x}{\sum^{\text{​}}\Delta t}$$
where *k* is the partition number of signal *X*, and A is the channel number, Δ*x* represents the difference between adjacent amplitudes of the EEG in one second and Δ*t* denotes the time interval:(3)$$\Delta x=x_{i+1}-x_{i}$$
(4)$$\Delta t=t_{i+1}-t_{i}$$
where *x*_i_ and *x*_i+1_ are the current and next EEG amplitude values, respectively, and *t_i_* and *t_i_*_+1_ represent the corresponding times i.The ΔEEG_A_ of each EEG channel is then computed as,(5)$${\Delta \mathrm{EEG}}_{A}=\frac{\sum_{i=1}^{K}{\Delta \mathrm{EEG}}_{A_{k}}\;}{K}$$

ΔEEG_A_ is the mean value for one EEG channel.

*Zero-crossing intervals (ZCI):* ZCI [30,85,130,131] is defined, in this context, as the time interval between a positive to negative voltage transition to the next positive to negative voltage transition (x-axis and y-axis were used for time and voltage representation, respectively). It is based on finding a set of instances when the waveform intersects with the time axis. The ZCI calculation for the N-sample EEG signal is obtained as,
(6)$$T=\{tif\;x_{t}>0\;and\;x_{t+1}<0\}\;$$
where *x_t_* and *x_t_*_+1_ are the times that EEG amplitude changed from positive to a negative value, respectively, and *T* is the vector that contains the time instances when the amplitude changed from positive to negative value (for example, *T* = {*t*_1_, *t*_2_, *t*_3_, …., *t_N_*}).
(7)$$\mathrm{ZCI}=\overset{K-1}{\underset{i=1}{\sum }}\Delta t\;$$
(8)$$\Delta t=t_{i+1}-t_{i}$$

ZCI is the zero-crossing interval value, *K* is the indicator for the number of instances, and ti and *t_i_*+1 represent the predecessor and successor corresponding to the instances.

*Tsallis entropy (TsEn):* The computation of TsEn [132] of an N-sample EEG data sequence, *x*(1), *x*(2), …, *x*(*N*), is based on the generalised measure of entropy due to Tsallis: (9)$${\mathrm{TsEn}}_{g}=\left(\overset{K}{\underset{i=1}{\sum }}P_{i}-P_{i}^{q}\right)/\left(q-1\right)$$
where TsEn_g_ is the Tsallis entropy value, *k* is the number of states that the amplitudes of the EEG are quantized into, *P_i_* is a probability associated with the *i*th state, and *q* is Tsallis parameter (*k* = 2200, and *q* = 0.5).

*Higuchi fractal dimension (HFD):* Higushi algorithm may be used to calculate the fractal dimension, *D_f_*, and complexity of time series such as the EEG [133]. The algorithm is based on a measure of the length of a curve, *L*(*k*), which represents the time series:*L(k)~K^−D^_f_*(10)
*D_f_* may be calculated by log-log curve fitting.

To compute the HFD biomarker [86,96,98,134] of an N-sample EEG data sequence *x*(1), *x*(2), …, *x*(*N*), the data is first divided into a *k*-length sub-data set as,
(11)$$X_{k}^{m}:x\left(m\right),x\left(m+k\right),x\left(m+2k\right),\ldots ,x\left(m+\left[\frac{N-m}{k}\right]\cdot k\right),$$
where [] is Gauss’ notation, k is constant, and *m* = 1, 2, …, *k*. The length *L_m_*(*k*) for each sub-data set is then computed as,
(12)$$L_{m}\left(k\right)=\left\{\left[\overset{\left[\frac{N-m}{k}\right]}{\underset{i=1}{\sum }}\;\left|x\left(m+ik\right)-x(m+\left(i-1\right)\Delta k\right|\right]\frac{N-1}{\left[\frac{N-m}{k}\right]\cdot k}\right\}/k$$

The mean of *L_m_*(*k*) is then computed to find the HFD for the data as,
(13)$$\mathrm{HFD}=\frac{1}{K}\overset{K}{\underset{M=1}{\sum }}\;L_{m}\left(k\right)$$

*Lempel-Ziv complexity (LZC):* To compute the LZC [75,99,100,101,103] biomarker of an N-sample EEG data sequence *x*(1), *x*(2), …, *x*(*N*), the EEG signal is first converted into a binary string as,
(14)$$x\left(i\right)=\{\begin{array}{c} 0\;\;if\;\;\mathrm{EEG}\left(i\right)<M\\ 1\;if\;\;\mathrm{EEG}\left(i\right)\geq M \end{array}$$
where *x*(*i*) is the equivalent binary value of EEG(*i*), i is the index of all values in the EEG signal, and *M* is the median value of each EEG channel. The median value is used to manage the outliers. The binary string is then scanned from left to right until the end to produce new substrings. A complexity counter *c*(*N*) is the number of new substrings. The upper bound of *c*(*N*) is used to normalise *c*(*N*) to get an independent value from the sequence of length *N*. The upper bound of *c*(*N*) is *N*/log2(*N*). c(*N*) is then normalised by *b*(*N*) as,
(15)$$C\left(N\right)=\frac{c\left(N\right)}{b\left(N\right)}$$
where *C*(*N*) is the normalised value of the LZC, and *b*(*N*) is the upper bound of the *c*(*N*). 

*Approximation entropy (ApEn):* ApEn [88,134,135,136] calculation of an N-samples EEG signal, two input constants (*m*, and *r*) must be identified to calculate ApEn that referred as ApEn (*m*, *r*, *N*), where m is the run length and r is the tolerance window. To calculate the ApEn, initialise the vector sequences *y*_1_, *y*_2_, …, *y*_(*N*-*m*+1)_, where *y*_i_ = [*x_i_*, *x_i_*_+1_, …. , *x_i_*_+m−1_], *I* = *1*, …, *N* − *m* + 1. These vectors represent m successive *x* values beginning with the *i*th point. Then, the distance is defined between *y_i_* and *y_j_* as the maximum differences between successive scalar values. For the *y_i_*, the *N^m^*(*i*) refers the number of *j* (*j* = 1, …, *N* − *m* + 1, *j* ≠ *i*) therefore, *d*[*x_i_*,*x_j_*] ≤ *r*. Therefore, for *i* = 1, …, *N* − *m* + 1:(16)$$C_{r}^{m}\left(i\right)=\frac{N^{m}\left(i\right)}{N-m+1}$$
where $C_{r}^{m}$ values compute the regularity within a tolerance r to the specified window *m*. Then, compute the average natural logarithm of each $C_{r}^{m}$ over *i*:(17)$$\phi^{m}\left(r\right)=\frac{1}{N-m+1}\overset{N-m+1}{\underset{M=1}{\sum }}\;lnC_{r}^{m}\left(i\right)$$

The dimension is increased to *m* + 1 and the previous steps will be repeated to get $C_{r}^{m+1}$ and $C_{r}^{m+1}$.

The final step the ApEn is defined as,
(18)$$\mathrm{ApEn}\left(m,r,N\right)=\phi^{m}\left(r\right)-\phi^{m+1}\left(r\right)$$

The ApEn value was computed for each channel.

*EEG Coherence (EEG-Coh):* Coherence [73,137] biomarker computation of an N-sample EEG data sequence *x*(1), *x*(2), …, *x*(*N*), is based on the coherence between two EEG channels and was calculated as,
(19)$$\mathrm{Coh}\left(a,b\right)=\frac{{\left|P_{a,b}\right|}^{2}}{PSD_{a}*PSD_{b}}$$
where *a*, and *b* are EEG channels, *PSDa*, and *PDSb* are the power spectral density for EEG channels *a* and *b*, |*P_a,b_*|^2^ is the square cross-spectral density of the channels *a*, and *b*. The EEG coherence is a value between 0 and 1 calculated using Welch’s power spectral density. It represents the functional relationships between two brain regions [73,138,139]. 

#### 2.2.2. Biomarker Selection and Construction of Panels (Steps 3 and 4)

The methodological framework involves a consideration of all possible biomarkers. This creates a large number of biomarkers at the biomarker computation stage. Therefore, it is necessary to select biomarkers with a statistically significant association with AD as these may be useful in discriminating between AD and Nold subjects. *p*-values, Bonferroni corrected for familywise error rate [140], and the probability distribution ratio (see later) were used for this purpose.

Biomarkers with p-values not greater than 0.001 are selected as having a significant association with AD [141,142]. Similarly, EEG channels for which the biomarkers have significant *p*-values (*p*-value < = 0.001) were selected as these are considered as having a significant association with AD. Figure 3 shows the procedure used to select the EEG biomarkers of AD. All possible combinations of the significant biomarkers are then used to construct panels of biomarkers. 

As an example, suppose we selected the following three biomarkers as having a significant association with AD:

TsEn(alpha(T3)), LZC(beta(T5)), and HFD(delta/theta(C4). Then, all the possible biomarker panels will have one biomarker, two biomarkers, and three biomarkers: One biomarker panels: [TsEn(alpha(T3))], [LZC(beta(T5))], and [HFD(delta/theta(C4))]Two biomarker panels: [TsEn(alpha(T3)), LZC(beta(T5))], [TsEn(alpha(T3)), HFD(delta/theta(C4))], and [LZC(beta(T5)), HFD(delta/theta(C4))]Three biomarker panels: [TsEn(alpha(T3)), LZC(beta(T5)), HFD(delta/theta(C4))]

In the study, Dataset B was used for biomarker selection and construction of panels.

#### 2.2.3. Diagnostic Model Development and Evaluation (Steps 5–7)

The outcome of biomarker selection is a set of biomarker panels from the different methods of computing biomarkers. At this stage, we want to determine how well the biomarker panels perform in detecting AD. The best performing biomarker panels are then used to develop a diagnostic model. We start by developing an SVM model for each biomarker panel using a 10-fold cross validation strategy. Dataset B was split into 60% for training and 40% for testing for this purpose. The models were further assessed in terms of sensitivity and specificity using the remaining 40% of dataset B.

Biomarker panels of models that obtained sensitivity and specificity of at least 80% were selected as the best performing panels. The selected biomarker panels are then combined to produce a diagnostic model using LDA classifier. The resulting model is then evaluated using dataset A.

#### 2.2.4. Optimisation and Validation of the Diagnostic Model (Steps 9 and 10)

Although the performance of the diagnostic model after Step 8 above is good, the number of EEG biomarkers involved may still be large. Further investigation is necessary to identify the smallest possible subset of EEG biomarkers from the panel of biomarkers that may be used in the diagnostic model to detect AD and still maintain high performance. At this stage, new and independent EEG datasets (C and D) are used in the investigation to avoid bias and overfitting. This also helps to assess how the diagnostic model would perform in different clinical settings. In this study, dataset D was used for training and cross-validation. Dataset C was used for subsequent testing to validate the optimized diagnostic model. 

The following steps outline the procedure for finding the smallest subset of biomarkers from the selected panel of biomarkers. 

As before, biomarkers in the final biomarker panels in Step 6 (see Section 2.2) are first computed using dataset D.New panels of biomarkers are then created from the biomarkers (i.e., by combining one or more biomarkers).For each of the new panel of biomarkers, develop and test a model to detect AD. In this study, SVM model was used to combine the biomarkers in each panel using a 10-fold cross validation strategy [123].Select panels and hence the subset of biomarkers that meet the diagnostic criteria and develop a diagnostic model from these.Validate the diagnostic model using unseen datasets (datasets A, B and C).

## 3. Results

### 3.1. Biomarker Computations

For each of the seven methods of biomarker computation (i.e., ΔPS, ΔEEG_A_, ZCI, TsEn, HFD, ApEn, and LZC) and for each channel, all the 25 possible biomarkers were computed (five biomarkers for the five EEG frequency bands and 20 biomarkers for the ratios between bands). Examples of the computed biomarkers are ZCI(alpha), TsEn(delta), TsEn(alpha/beta), and ΔEEG_A_(theta/beta). Thus, 475 biomarkers were computed and analysed for each of the seven methods (25 biomarkers × 19 EEG channels for each method). Thus, there were a total of 3325 biomarkers were computed for the seven indices (475 × 7). For the coherence based approach, there were 171 coherence values between pairs of the 19 EEG channels (e.g., Fp1-Fp2, Fp1-F7, …, O1-O2) were computed for each frequency band and 20 ratios. The total number of biomarkers computed for EEG coherence was 4275 (171 coherence values × 25 biomarkers). A total of 7600 (3325 + 4275) single biomarkers were computed from the eight methods (i.e., ΔPS, ΔEEG_A_, ZCI, TsEn, HFD, ApEn, LZC, and EEG coherence). 

### 3.2. Biomarker Selection and Biomarker Panels

The *p*-values were computed for each of the 7600 biomarkers as they reflect the significance of the biomarker in AD detection. 

As an example, Table 2 shows the *p*-values and the Bonferroni corrected *p*-values [140] for the theta/alpha biomarker for the TsEn method. The critical *p*-value is 0.00263 (e.g., 0.05/19). This is found by dividing the familywise error rate (i.e., 0.05) by the number of tests, i.e., 0.05/N [140].

As can be seen, only two EEG channels (T5 and T6) are significant as they have Bonferroni corrected p-values of less than the critical *p*-value (i.e., 0.00263), and they also satisfy the *p*-value selection criterion i.e., *p*-value ≤ 0.001.

To assess the relative effects of the EEG features and EEG channels in the detection of AD, the probability distribution ratios [143] were computed as,
(20)$$P_{x_{i}}=\frac{x_{i}}{\sum_{i=1}^{N}X_{i}}$$
where *P_Xi_* is the probability distribution ratio for an N-sample data sequence *x_i_*, *x_i_*_+1_, …, *x_N_*, *I* = 1… *N*.

Table 3 summarises the EEG features and the number of EEG channels for which the p-value criterion was satisfied for each method of biomarker computation (i.e., ΔPS, ΔEEG_A_, ZCI, TsEn, HFD, ApEn, and LZC). For example, for the HFD method of biomarker computation in the alpha band, the p-value criterion was satisfied at 4 EEG channels 

To assess the impact of each biomarker feature in AD detection, the total number of channels at which the p-value criterion is satisfied, and the probability distribution ratio were computed for each biomarker feature (see last three columns of Table 3) and used to rank the biomarker features. As shown in Table 3, the theta/alpha ratio has the highest number of channels that have a *p*-value which is less than or equal to 0.001. While delta/gamma and beta/gamma biomarker features have no EEG channels that satisfied the threshold of the *p*-value. 

The maximum probability distribution ratio was 15.579 for the theta/alpha ratio. This was computed by dividing the probability distribution ratio of theta/alpha i.e., 74 by the total probability distribution ratio of all bands i.e., 475, and multiplying the resulted value by 100. 

To determine the EEG biomarker features that have a significant association with AD biomarker features that have cumulative probability distribution ratio of 80% or more were selected (i.e., theta/alpha, alpha/theta, alpha/delta, beta/theta, theta/beta, alpha, delta/alpha, delta, theta, theta/delta, and delta/theta in Table 3). These 11 biomarker features have the greatest association with AD and are used to construct biomarker panels e.g., [LZC (alpha(T3)), LZC (delta/theta(P4)]. 

To determine the EEG channels that have a significant association with AD, we selected the channels that met the *p*-value threshold of less than or equal to 0.001. Then, the probability distribution ratio was also computed for all the 19 EEG channels to identify the EEG channels that are most promising in terms of significant association with AD. 

Table 4 shows the number of EEG channels for the methods (i.e., PS, ΔEEG_A_, ZCI, TsEn, HFD, ApEn, and LZC) that met the p-value threshold and the probability distribution ratio for each channel. The EEG channels are ranked in terms of probability distribution ratio and the total number of biomarker features. 

As can be seen, channel P4 was the top-ranked channel across the methods and features, and channels P3 and PZ were the second highest ranked, whilst channel Fp2 is the lowest ranked. Similar findings have been reported in the literature in which AD is thought to originate from the back of the brain and then spread to other areas of the brain [85,97,99,100,101,103]. This is consistent with the spatio-temporal pattern of AD-related degeneration, where hippocampal atrophy, the earliest sign of the disease, is followed by a widespread medial temporal lobe volume loss [144], which in turn gradually spreads through parietal, frontal, and temporal cortices until the whole-brain atrophy is observed at the latest stages [145]. Furthermore, the distribution of tau pathology in the brain, whereas recent amyloid-PET studies have shown that amyloid pathology, an “initial insult,” starts in association cortices and spreads from neocortex to allocortex [146].

Based on the analysis of the results (see Table 3 and Table 4), 12 EEG channels (P4, P3, PZ, T6, T5, C4, T3, C3, CZ, T4, O2, and O1) were selected as having a significant association with AD. These channels accounted for more than 80% of the cumulative probability distribution ratio of all EEG channels, as shown in Table 4. 

The 12 EEG channels selected have been reported in other studies as having a significant role in AD detection [101,147,148,149,150]. 

For each of the 11 EEG biomarkers that were selected, 4082 combinations were investigated for the 12-EEG channels (all combinations from length 1 to length 10 were constructed for the 12-EEG channels). For each method, we investigated 44,902 biomarkers (11-biomarkers × 4082-combinations).

Following a similar analysis for biomarkers based on EEG coherence, 10 pairs of EEG channels that satisfied the p-value threshold (less than or equal to 0.001) were selected, Table 5.

Panels of biomarkers with the selected features and EEG channels were created for each of the seven methods (ΔEEG_A_, ΔPS, ZCI, TsEn, HFD, ApEn, LZC). Similarly, panels of biomarkers for the coherence method were also created. Altogether, a total of 325,567 biomarker panels were created. 

An SVM model for each panel in each method was developed and its performance assessed using dataset B (dataset B was split into 60% for training and 40% for testing for this purpose). The best biomarker panels for each method were determined based on their performance (in terms of sensitivity, specificity, and the number of EEG channels used). Sensitivity and specificity thresholds of at least 80% in the detection of AD were set for the biomarkers. Table 6 summaries the features of the best performing biomarker panels. For example, in Table 6, the third biomarker panel for the ApEn method is Alpha (P3, T6, O1). This biomarker panel is based on three features—alpha features derived from channels P3, T6 and O1) 

### 3.3. Diagnostic Model to Detect AD

A key goal in the study is to find the best combination of EEG based biomarkers to detect AD patients with high sensitivity and specificity. Biomarker panels that satisfied the threshold for sensitivity and specificity of at least 80% and have the fewest number of EEG channels were selected for inclusion in the diagnostic model. Table 7 summaries the best 17 biomarker panels selected on the basis of this. As can be seen from the table, the 17 biomarker panels contain a total of 69 panels of biomarkers (69 out of 325,567 panels of biomarkers). These 69 panels of EEG biomarkers were then combined in a model using LDA classifier. The combined biomarker consists of 30 panels of biomarkers from analysis of the reduction in EEG complexity, 33 panels of biomarkers from analysis of the slowing of the EEG, and 6 biomarkers panels of from the analysis of the decrease in functional connectivity of EEG. The biomarkers panels were combined in a one diagnostic model.

The training and testing EEG datasets were used in the model development (dataset B was split into 60% for training and 40% for testing for this purpose). While dataset A was used to evaluate the model and the performance found to be 100% for sensitivity and specificity. The performance of the diagnostic models based on the individual panel biomarkers and the whole 17 panels are summarised in Table 7. As can be seen, the combined model outperforms the individual panels.

Although the performance of the combined model is good, the model may not be using the smallest possible subset of EEG biomarkers. 

To show the effect of AD on EEG signal. The changes in EEG due to AD can be shown for the 69 robust EEG biomarkers that may have a more significant association with AD. Table 8 shows the changes in EEG signal due to AD for the 17 EEG biomarkers (e.g., ApEn(Alpha), ApEn(Alpha/Theta)) that may have a more significant association with AD for the three key characteristics of dementia EEG (i.e. slowing of the EEG, reduction in EEG complexity and reduction in EEG connectivity). The shaded boxes in Table 8 referred to the decrease in EEG biomarkers due to AD, otherwise, it means the increase. Table 8 shows the changes in EEG characteristics due to AD.

Table 7 shows, that EEG coherence provides less performance compared to the complexity and slowing of EEG and may be due to inflation caused by volume conduction [151]. The effects of this may be reduced using Surface Laplacian approach [152].

### 3.4. Optimisation and Validation of the Diagnostic Model 

The selected biomarker panels at Step 6 above contains a total of 69 biomarkers. Starting with the 69 biomarkers, finding the smallest subset of biomarkers involves creating new panels of biomarkers by combining the biomarkers. In this study, the maximum number of biomarkers in a new panel was limited to four because the goal is to find the smallest subset of biomarkers (i.e., panels with the fewest number of biomarkers) to detect AD with acceptable performance and the need to avoid the exponential increase in the number of possible biomarker panels as panel size increases. Limiting the panel size to four still yielded 919,310 biomarker panels (see Table 9). Dataset D was used to compute the 919,310 EEG biomarker panels.

A machine learning model was developed for each biomarker panel in the table above using a 10-fold cross validation. Single-biomarker panels, two-biomarker panels, three-biomarker panels, and four-biomarker panels that satisfy the performance threshold (sensitivity and specificity equal to or greater than 80%) were identified.

To select the best biomarkers from these biomarker panels, we focused on the smallest subset of biomarkers that have a high performance in AD detection. Based on this, we found that the following six biomarker panels have the best performance:

TsEn(alpha/theta(T6)), ZCI(alpha/theta(P3)), ZCI(delta/alpha(P3)), ΔEEG_A_(alpha/delta(T6)), ΔEEG_A_(theta/alpha(T3)), and ΔEEG_A_(theta/alpha(T5)).

The six biomarkers were then combined using an SVM model. Dataset D using 10-fold cross-validation was used in the development of the final model. 

It is to be noted that of the eight methods of computing EEG biomarkers, only three appear in the final model—TsEn, ZCI and ΔEEG_A_, and that only alpha and theta features are in the model. ΔEEG_A_ shows promising results in quantifying the slowing of EEG in AD detection [23]. ZCI is a promising method to quantify changes in the EEG due to AD [30,85]. TsEn is one of the most promising information theoretic methods for quantifying EEG complexity [48,61,84,92,93]. Its fast computation may serve as the basis for real-time decision support tools for diagnosing AD [48,84,92].

The resulting model was tested and validated using unseen datasets. In particular, datasets A, B, and C were used to validate the developed model. The performance of the final model using dataset C was 100% for sensitivity and specificity, respectively. For dataset A, it was also 100% for sensitivity and specificity, respectively. For dataset B, it was 85% for sensitivity and 100% for specificity. 

### 3.5. The Effect of Combining Multiple Biomarkers on Performance

To assess the effect of combining multiple biomarkers to produce a robust biomarker, TsEn method was selected as an example of the eight methods investigated (i.e., ApEn, LZC, HFD, TsEn, ΔPS, ΔEEG_A_, ZCI, and coherence). Combining biomarkers was performed in developing the diagnostic models. To demonstrate how the biomarkers were combined, for example, the six biomarkers [TsEn(alpha/theta(T6)), ZCI(alpha/theta(P3)), ZCI(delta/alpha(P3)), ΔEEG_A_(alpha/delta(T6)), ΔEEG_A_(theta/alpha(T3)), and ΔEEG_A_(theta/alpha(T5))] were combined during the training of diagnostic model that contains the values of these six biomarkers belong to Nold and AD, then test the developed model using the same biomarkers but for an unseen dataset. Table 10 shows the performance of the alpha band for TsEn method. The table shows the effect of combining multiple EEG biomarkers in a diagnostic model to produce a new biomarker that has a higher performance than its elements.

As shown in Table 10 the performance of [TsEn(alpha(C3)), TsEn(alpha(P3))] was 88.33% and 81.82% for sensitivity and specificity, respectively. The performance of TsEn(alpha(C3)) was 50% for sensitivity and 66.67% for specificity and the performance of TsEn(alpha(P3)) was 35.71% for sensitivity and 33.33% for specificity. Also, the performance of [TsEn(alpha(C3)), TsEn(alpha(O2))] is higher than the performance of TsEn(alpha(C3)) and TsEn(alpha(O2)) separately. This finding is consistent with the results in [3,23,34,40,108], they illustrate the point that the performance of combining multiple biomarkers to produce a composite biomarker outperforms single biomarkers alone.

As we have already seen, the final diagnostic model which combines six biomarkers [TsEn(alpha/theta(T6)), ZCI(alpha/theta(P3)), ZCI(delta/alpha(P3)), ΔEEG_A_(alpha/delta(T6)), ΔEEG_A_(theta/alpha(T3)), and ΔEEG_A_(theta/alpha(T5))] has a performance of 100% for sensitivity and specificity for datasets A and C and 85% for sensitivity and 100% for specificity for dataset B. 

Table 11 demonstrates using multiple EEG biomarkers in one diagnostic model. These biomarker panels include one, two, three, four, five, and six biomarkers. For example, the panel (e.g., [ΔEEG_A_(alpha/delta(T6))]) means the developed diagnostic model has one biomarker, the panel (e.g., [ZCI(alpha/theta(P3)), ΔEEG_A_(theta/alpha(T3))]) means the developed diagnostic model has two biomarkers. As can be seen in Table 11, combining multiple biomarkers of different techniques e.g., slowing of EEG and EEG complexity provides better performance than using each biomarker separately. This finding is consistent with other findings that suggest combining EEG biomarkers could be more sensitive to disease progression, identify optimal combinations of biomarkers, and could complement other AD biomarkers e.g., PET, CSF and MRI [34,110]. The combining of multiple biomarkers can reveal hidden separation boundaries [110].

## 4. Discussion 

Many EEG-derived biomarkers have been reported as promising in the detection of AD in the literature [55,58,111,153,154,155]. Here we tested the hypothesis that by combining complementary information obtained with different EEG biomarkers across multiple frequency bands with different analytical techniques the detection performance can be increased. AD affects cognitive memory and brain functions [3,156] and each EEG frequency band is associated with specific brain functions [104,156]. Thus, a decline in brain functions may be reflected in the EEG activities and EEG frequency bands [43,44,45]. Consequently, deriving EEG biomarkers from frequency bands is thought to provide a better performance in detecting AD compared to EEG biomarkers derived from the whole EEG record [55]. 

The development of robust EEG based biomarkers requires a thorough investigation of all the possible factors that affect AD detection. These factors may include the basis for the biomarkers (e.g., EEG slowing, reduction in EEG complexity, decrease in EEG connectivity), the EEG features used and hence the methods used to derive the biomarkers, and the EEG channels from which the biomarker is derived. Taking these factors into account and integrating different biomarkers into one biomarker should lead to robust EEG based biomarkers. Our methodological framework makes this possible and our results support this.

There is an ongoing development in the field of EEG biomarkers of neurodegenerative disorders (i.e., dementia with Lewy bodies, Parkinson’s disease), suggesting that EEG recordings might support distinguishing between different forms of dementia, even before the onset of distinctive clinical symptoms [157]. Therefore, the application of a developed analytical framework to other neurodegenerative conditions might provide a reproducible and standardized workflow for the development of robust EEG biomarkers with good discriminating performance.

The findings in this study confirmed that EEG could provide good biomarkers to detect AD. Table 6 and Table 7 show the best biomarkers that can be used to detect AD. These biomarker features have been reported in previous studies to be important in AD detection [56,63,154,158,159,160,161,162,163,164,165,166,167,168,169,170,171,172,173]. For example, Fahimi et al. [169] and Schmidt et al. [167] found that theta/alpha and alpha/theta are good biomarkers for AD detection and this is consistent with the finding in Table 7. The HFD (Theta/alpha (T3, T5)) and ΔEEG_A_ (Theta/alpha (T3, C3, T5, P3, PZ, P4)) biomarkers provided high performance in AD detection. González-Castro et al. [170] and Arns et al. [171] used beta/theta ratio to diagnose attention deficit hyperactivity disorder (ADHD), and Barry et al. [172] reported that theta/beta ratio is a marker of ADHD. While Zhang et al. [173] indicated that AD patients show increased ADHD symptoms. Thus, our findings suggest that beta/theta and theta/beta ratios may be important in AD detection. Reduction in alpha band activity is reported to be an indicator of MCI and was observed in progressive stages of AD [63,158]. Our finding in Table 6 shows that alpha provided the best performance to detect AD. Using the delta/alpha ratio, Ladurner et al. [159] found that Ischaemic stroke (IS) disorder is common in dementia. Besides, Finnigan et al. [160] concluded that the delta/alpha ratio is an optimal QEEG (Quantitative Electroencephalogram) index for IS patients. Morettiet et al. [161], Hier et al. [162], and Musaeus et al. [163] found that features in the delta band are a marker to discriminate between AD patients and Nold subjects. For alpha and theta bands, alpha band was significantly lower in AD patients [56,154]. Also, Sutter et al. [164] found that the theta band was associated with brain atrophy [164]. As shown in Table 6, LZC (Alpha (C3, CZ, PZ, T6, O1)), ApEn (Alpha (P3, T6, O1)), ZCI (Alpha(T6, O1), and ZCI(Alpha(T6, O2)) provided the best performance in AD detection. For theta/delta, Sutter et al. [164] found that theta/delta of EEG is associated with intracerebral hemorrhages (ICH). Moreover, Cohen et al. [165] indicated that the more pathogenic form of Aβ_1-42_ was found to be highly associated with ICH [165]. Wacker et al. [166] found that VAL allele is associated with an increased delta/theta ratio of EEG. According to Ventriglia et al [168], there was a substantial increase in the number of individuals carrying two copies of the Val allele in AD patients compared to healthy controls. Results in Table 6 show the ΔPS (Theta/delta(P3, T6)) provided the best performance to detect AD.

We selected EEG biomarkers and channels with a significant association with AD detection as having high performance in distinguishing between AD patients and Nold subjects. These biomarkers and EEG channels were then used to find the best combination of the EEG biomarkers that is robust enough to be used to detect AD. In this study, the EEG biomarkers have performed well, suggesting that the methodology is realistic for AD detection. 

For clinical acceptance, there is a need for further studies using larger datasets from different settings to assess the full potential of the model and methodology. Finding the smallest subset of biomarkers from 69 biomarkers involves examining an excessively large number of biomarker panels. As a result, the maximum number of biomarkers in a panel was limited to four and the determination of the smallest subset was based on these. Thus, the subset we used to develop the final diagnostic model may not be the optimal subset (in terms of performance, number and type of biomarkers). In future, it should be possible to explore other approaches to finding optimum subsets of features.

The selected biomarkers included one biomarker based on the reduction in EEG complexity i.e., TsEn(Alpha/theta(T6)), and five biomarkers based on the slowing of the EEG i.e., ZCI(Alpha/theta(P3)), ZCI(Delta/alpha(P3)), ΔEEG_A_(Alpha/delta(T6)), ΔEEG_A_(Theta/alpha(T3)), and ΔEEG_A_(Theta/alpha(T5)). Furthermore, analysis of the results shows that the EEG channels in the temporal and parietal lobes (i.e., T6, T3, T5, and P3) gave better results compared to other channels that relate to other lobes e.g., the frontal lobe. 

Results of the present study are consistent with the previous observations of abnormal cortical EEG and magnetoencephalographic (MEG) rhythms in parietooccipital and temporal channels [58,174,175,176,177]. The most consistently reported abnormalities of cortical rhythms in AD patients are an increase of delta and theta activity and a reduction of alpha activity [178,179,180,181,182,183,184]. In line with these observations, an earlier classification study has reached the highest discrimination accuracy between Nold and AD participants using composite markers of source current density in parietal, temporal, and occipital regions at alpha, theta, and delta frequency bands [185]. Together with previous findings, the present study supports the important role of temporoparietal sources of EEG activity at alpha, theta, and delta frequencies in discrimination between patients with AD and Nold participants.

What is the neurophysiological role of these changes in the spectral composition of EEG rhythms in AD? In quiet wakefulness, scalp rsEEG rhythms at posterior channels are dominated by alpha-band oscillations. These oscillations reflect the fluctuating cortical inhibition and the underlying widespread synchronization of cortical pyramidal neurons [186]. Indeed, studies have shown that neural signals synchronized at around 10 Hz subserve vigilance and attention [45,70,187,188]. The increased theta and delta rhythms, in turn, have been linked to sleep, fatigue, and decreased alertness (drowsiness) [189,190]. EEG markers computed at these frequencies have proven as reliable indicators of neural inhibition and impaired information processing [191]. Thus, computing frequency ratios between delta/theta and alpha rhythms allow to integrate information embedded into abnormalities at lower and higher frequencies of the spectral envelope into a single composite biomarker, indicative of cortical neural synchronization. 

Although the performance of the diagnostic model is good, there are a number of significant limitations of the study. First, the size of the dataset used in the model development and testing of the model is small. In the study, 40 cases were used for training the model (i.e., 20 AD patients and 20 Nold subjects to dataset D), and 72 cases for testing (i.e., 30 AD patients and 42 Nold subjects to datasets A, B, and C). Although this compares well with the size of the dataset in other studies (e.g., Amezquita-Sanchez et al. [192] used 74 cases, 37 MCI and 37 AD patients with an accuracy of 90.3%. Chai et al. [193] used 20 cases, 10 AD patients and 10 healthy people with AUC reaching 0.89), the number of subjects is still quite low and there is a risk of over-fitting.

As mentioned in the Introduction section, progressive neurodegeneration caused by AD pathology manifests in detrimental and gradual changes in cognition (e.g., cognitive impairment and memory decline), brain structure (e.g., medial-temporal lobe and hippocampal atrophy) and function (e.g., altered brain metabolism, resting state brain networks and EEG signals). For the time being, the available treatment options for patients diagnosed with AD dementia can provide only temporal symptomatic relief. Improved treatment methods are now under investigation; however, it is a widely accepted opinion, that to be effective they have to be administrated at the early stages of the disease. To detect patients who are going to develop AD dementia in future, a large number of cognitively normal elderly individuals have to undergo preventive screening. This scenario requires biomarkers that are capable of detecting early pathological changes but also non-invasive and can be obtained at a low cost. 

Although EEG is not included in the current diagnostic guidelines for AD and not endorsed for use in clinical trials performed in AD patients [41], EEG is fundamental to studying the neuronal and synaptic loss caused by the progression of the disease [194]. Whereas the most recently accepted biomarkers of AD predominantly reflect AD-related molecular and structural brain changes, they give no information about functional deficits in patients with AD. Given a surge in recent evidence that synaptic loss and dysfunction play a key role in AD pathogenesis, markers that measure deficits of brain function and provide insights into brain synaptic activity are of great potential. Here comes the rsEEG (resting-state electroencephalographic)—a widely available, cost-efficient, and non-invasive technique able to highlight the functional changes in electrical activity generated by postsynaptic potentials from cortical neurons. Indeed, studies repeatedly show that abnormal EEG markers relate to cognitive deficits in AD patients at different disease stages. These converging findings support the application of the EEG markers as screening biomarker in population studies and drug discovery [109].

In the last decade, the research definition of AD has moved from a clinical to a more biological paradigm [195]. In this contexts diagnosis of AD is based on biomarkers sensitive to amyloidosis (decreased Aβ42 in the CSF and increased retention of amyloid tracers on PET), tauopathy (increased tau and phospho-tau in CSF) and neurodegeneration (hippocampal atrophy on MRI, cortical hypo-metabolism on 18FDG-PET). Although these biomarkers capture relevant aspects of AD pathology, their application on clinical and preclinical population is still limited [196] due to their high cost and invasiveness (which is especially true for studies requiring longitudinal monitoring of patients, such as clinical trials) [41]. In this respect, the present EEG procedures may provide information about AD-related neurophysiological alterations related to oscillatory synchronization/desynchronization and coupling/decoupling of cortical neuronal activity in agreement with recommendations of the International Society to Advance Alzheimer’s Research and Treatment (ISTAART) Electrophysiology Professional Interest Area (EPIA) [111], These procedures may be exploited in monitoring the development of the disease over years and trace response to candidate disease modifying drugs in clinical trials. 

## 5. Conclusions

AD-related neurodegeneration causes alterations in synaptic and neural activity. These alterations might be reflected in the information content of the EEG signals. Thus, EEG provides valuable information about changes in electrophysiological brain dynamics due to AD and this may be used to detect AD.

This study provides a novel framework for constructing robust biomarkers that can be used to detect AD with high performance (sensitivity and specificity closed to 100%) by exploiting the combined strengths of different biomarkers. The resulting EEG biomarkers may be used in clinical studies performed in patients with AD in response to the need for a gatekeeper screening tests of large groups of the aging population. 

The main limitation of this study was the size of the cross-sectional dataset of AD patients and Nold subjects. In addition, specificity and association with established biomarkers such as CSF tau or PET amyloid imaging should be investigated. For future work, we will evaluate our method with larger size longitudinal EEG datasets, from different settings, that contain normal, MCI and AD subjects.

## Figures and Tables

**Figure 1 brainsci-11-01026-f001:**
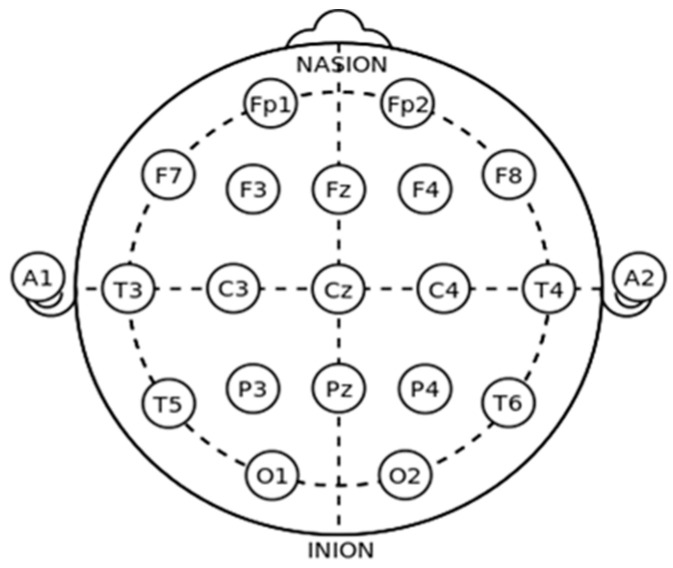
International 10–20 system [114].

**Figure 2 brainsci-11-01026-f002:**
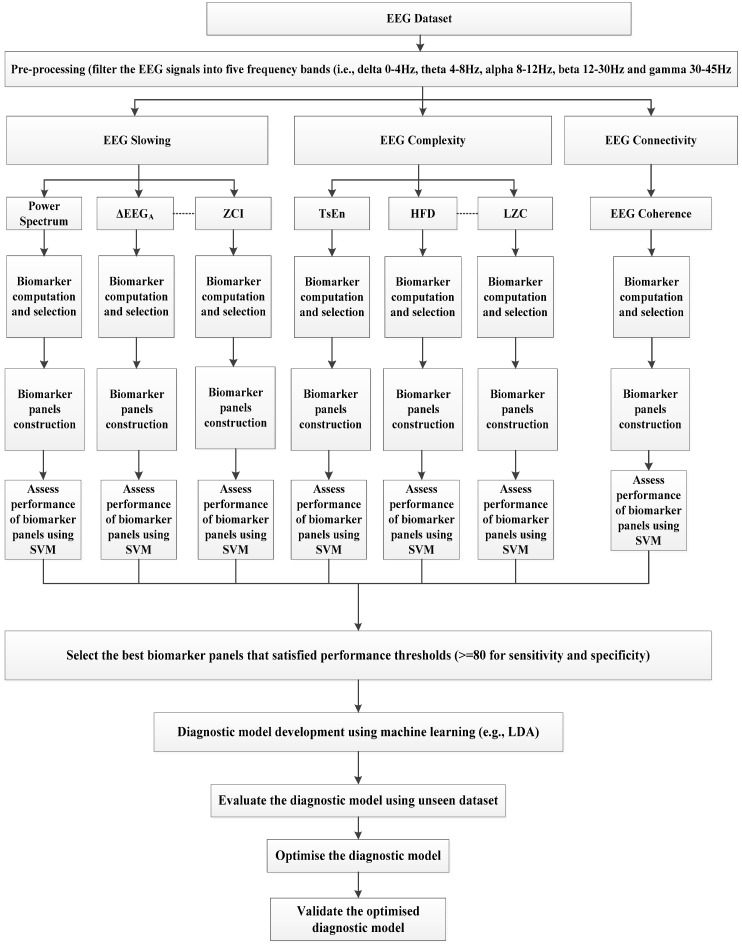
Methodological framework that was used for developing robust EEG based biomarkers.

**Figure 3 brainsci-11-01026-f003:**
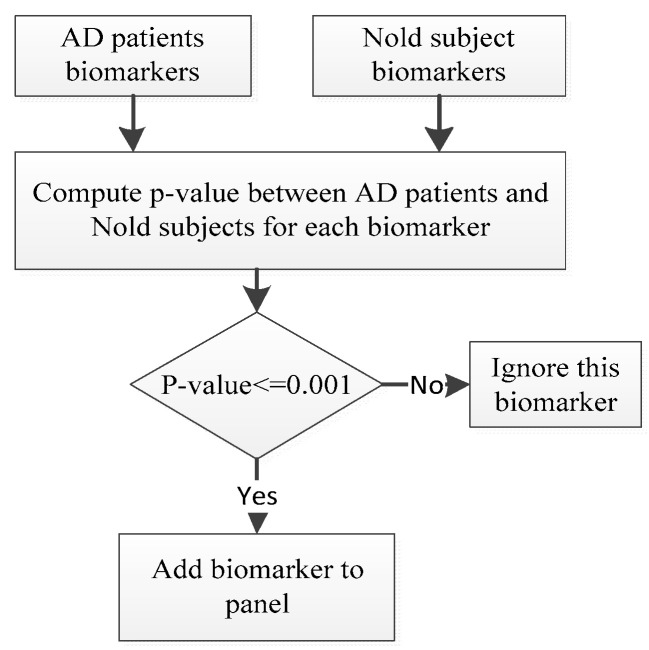
Construct panels of biomarkers for AD detection.

**Table 1 brainsci-11-01026-t001:** Description of the EEG dataset that were used in this study.

Dataset	N (AD/Nold)	Age		Male/Female		MMSE	
		AD [SD] (Years)	Nold [SD] (Years)	Nold	AD	Nold	AD
A	11(3/8)	>65 years	>65 years	-	-	-	-
B	41(17/24)	77.6 ± 10.0	69.4 ± 11.5	10/14	9/8	-	-
C	20(10/10)	78.3 ± 4.03	78.0 ± 4.24	5/5	5/5	29.19 ± 0.61	22.81 ± 2.37
D	40(20/20)	77.8 ± 5.50	75.75 ± 3.85	10/10	10/10	28.35 ± 1.00	21.21 ± 3.13

**Table 2 brainsci-11-01026-t002:** *p*-value for theta/alpha ratio for TsEn method.

Channel No.	EEG Channel	*p*-Value	Bonferroni-Corrected *p*-Value	Bonferroni-Corrected Significance
1	Fp1	0.0264	0.5016	not significant
2	Fp2	0.0394	0.7486	not significant
3	F7	0.4883	1	not significant
4	F3	0.5511	1	not significant
5	FZ	0.3612	1	not significant
6	F4	0.1582	1	not significant
7	F8	0.3105	1	not significant
8	T3	0.1352	1	not significant
9	C3	0.2859	1	not significant
10	CZ	0.8901	1	not significant
11	C4	0.5549	1	not significant
12	T4	0.2418	1	not significant
13	T5	0.0010	0.019	significant
14	P3	0.0048	0.0912	not significant
15	PZ	0.0207	0.3933	not significant
16	P4	0.0036	0.0684	not significant
**17**	**T6**	**0.0002**	**0.0038**	**significant**
18	O1	0.0028	0.0532	not significant
19	O2	0.0035	0.0665	not significant

**Table 3 brainsci-11-01026-t003:** Probability distribution ratio for all 25 EEG features for each method.

EEG Features	Number of EEG Channels That Meet *p*-Value Criterion for Each Method								Total Number of EEG Channels	Probability Distribution Ratio %	Cumulative Probability Distribution Ratio %
	ApEn	LZC	HFD	TsEn	ΔPS	ΔEEG_A_	ZCI	Coh			
**Theta/alpha**	**12**	**5**	**9**	**2**	**18**	**12**	**15**	**1**	**74**	**15.579**	**15.579**
**Alpha/theta**	**12**	**7**	**0**	**2**	**17**	**15**	**15**	**0**	**68**	**14.316**	**29.895**
**Alpha/delta**	**0**	**0**	**0**	**0**	**18**	**11**	**8**	**0**	**37**	**7.789**	**37.684**
**Beta/theta**	**13**	**0**	**0**	**0**	**2**	**10**	**9**	**1**	**35**	**7.368**	**45.052**
**Theta/beta**	**12**	**0**	**3**	**0**	**4**	**5**	**9**	**0**	**33**	**6.947**	**51.999**
**Alpha**	**2**	**0**	**4**	**0**	**19**	**3**	**4**	**0**	**32**	**6.737**	**58.736**
**Delta/alpha**	**1**	**0**	**0**	**0**	**18**	**4**	**7**	**0**	**30**	**6.316**	**65.052**
**Delta**	**0**	**0**	**0**	**0**	**19**	**0**	**0**	**3**	**22**	**4.632**	**69.684**
**Theta**	**0**	**0**	**0**	**0**	**18**	**0**	**0**	**3**	**19**	**4.000**	**73.684**
**Theta/delta**	**0**	**0**	**0**	**0**	**17**	**0**	**0**	**0**	**18**	**3.789**	**77.473**
**Delta/theta**	**8**	**0**	**0**	**1**	**0**	**4**	**4**	**0**	**17**	**3.579**	**81.052**
Gamma/theta	0	0	0	0	16	0	0	0	17	3.579	84.631
Beta/delta	0	0	0	0	2	7	3	2	14	2.947	87.578
Beta	2	0	4	0	1	1	3	2	13	2.737	90.315
Theta/gamma	6	0	0	1	0	0	4	0	11	2.316	92.631
Alpha/beta	0	4	0	0	4	0	0	0	8	1.684	94.315
Gamma/delta	3	0	0	0	1	3	1	0	8	1.684	95.999
Delta/beta	0	0	0	0	2	0	2	0	4	0.842	96.841
Alpha/gamma	0	4	0	0	0	0	0	0	4	0.842	97.683
Gamma	0	2	0	0	1	0	0	2	3	0.632	98.315
Gamma/alpha	0	0	1	0	2	0	0	0	3	0.632	98.947
Gamma/beta	0	1	0	0	1	0	0	0	3	0.632	99.579
Beta/alpha	0	0	0	0	1	0	0	0	2	0.421	100
Delta/gamma	0	0	0	0	0	0	0	0	0	0.000	100
Beta/gamma	0	0	0	0	0	0	0	0	0	0.000	100
Summation (Sum)	71	23	21	6	181	75	84	14	475	100	

**Table 4 brainsci-11-01026-t004:** Probability distribution ratios for all 25 EEG biomarker features and for all 19 EEG channels.

EEG Channels	Number of Biomarkers							Total Number of EEG Biomarker Features	Probability Distribution Ratio %	Cumulative Probability Distribution Ratio %
	ApEn	LZC	HFD	TsEn	ΔPS	ΔEEG_A_	ZCI			
**P4**	**7**	**5**	**4**	**0**	**10**	**10**	**12**	**48**	**10.412**	**10.412**
**P3**	**9**	**2**	**4**	**0**	**9**	**8**	**13**	**45**	**9.761**	**20.173**
**PZ**	**8**	**2**	**3**	**2**	**11**	**9**	**10**	**45**	**9.761**	**29.934**
**T6**	**3**	**6**	**2**	**2**	**9**	**7**	**7**	**36**	**7.809**	**37.743**
**T5**	**2**	**3**	**4**	**2**	**9**	**6**	**7**	**33**	**7.158**	**44.901**
**C4**	**6**	**0**	**0**	**0**	**11**	**6**	**4**	**27**	**5.857**	**50.758**
**T4**	**4**	**2**	**1**	**0**	**8**	**7**	**3**	**25**	**5.423**	**56.181**
**T3**	**4**	**0**	**1**	**0**	**14**	**3**	**2**	**24**	**5.206**	**61.387**
**C3**	**4**	**0**	**0**	**0**	**12**	**4**	**4**	**24**	**5.206**	**66.593**
**CZ**	**6**	**0**	**0**	**0**	**9**	**3**	**6**	**24**	**5.206**	**71.799**
**O2**	**3**	**3**	**1**	**0**	**10**	**3**	**4**	**24**	**5.206**	**77.005**
**O1**	**4**	**0**	**1**	**0**	**8**	**3**	**4**	**20**	**4.338**	**81.343**
F8	2	0	0	0	12	1	2	17	3.688	85.031
F7	3	0	0	0	9	2	2	16	3.472	88.503
F3	3	0	0	0	9	1	2	15	3.255	91.758
FZ	3	0	0	0	7	2	2	14	3.037	94.795
Fp1	0	0	0	0	9	0	0	9	1.952	96.747
F4	0	0	0	0	9	0	0	9	1.952	98.699
Fp2	0	0	0	0	6	0	0	6	1.302	100.001
Total	71	23	21	6	181	75	84	461	100	

**Table 5 brainsci-11-01026-t005:** Probability distribution ratio to EEG coherence for all 25 EEG biomarkers and all 19 EEG channels.

EEG Channel Pair	No. of Biomarkers	Probability Distribution Ratio	Cumulative Probability Distribution Ratio %
F4-F8	3	21.429	21.429
Fz-F8	2	14.286	35.714
T3-T4	2	14.286	50.000
Fp2-F4	1	7.143	57.143
F4-T3	1	7.143	64.286
F4-T4	1	7.143	71.429
F8-P4	1	7.143	78.572
T3-P4	1	7.143	85.715
T3-T6	1	7.143	92.857
T4-P3	1	7.143	100.000
Total	14	100.000	

**Table 6 brainsci-11-01026-t006:** The best performance of EEG biomarkers for the investigated eight indices.

Analysis Method	EEG Biomarker	EEG Channels
ApEn	Alpha/theta	(C4)
	Theta/alpha	(C4)
	Alpha	(P3, T6, O1)
LZC	Alpha/theta	(T3, T6)
	Theta/alpha	(T3, T6)
	Alpha/delta	(CZ, C4, P3, P4), (CZ, T4, P3, P4), and (CZ)
	Beta/theta	(C4, P3, T6, O1), and (C4, P3, T6, O2)
	Alpha	(C3, CZ, PZ, T6, O1)
	Delta/alpha	(CZ, T4, P3, P4), (CZ, T4, P3, T6), and (CZ, T4, P3, O2)
	Theta	(T3, C3, P3)
HFD	Theta/alpha	(T3), (T5)
	Alpha/delta	(CZ, PZ, O2)
	Alpha	(T5, PZ), (PZ, T6), (T6, O1) and (T6, O2)
TsEn	Alpha/theta	(T3, CZ, P3, T6), and (C3, CZ, O1, O2)
	Theta/alpha	(T3, CZ, C4, T6, O1), and (T3, CZ, T5, P4, O1)
	Alpha/delta	(C3, T6)
	Delta/alpha	(C3, T6)
ΔPS	Alpha/theta	(T3), (C3), (T4), (P3), (PZ), and (P4)
	Theta/alpha	(C3), (CZ), (C4), (P4), and (O1)
	Alpha/delta	(T3, T4, P3, P4, O1)
	Beta/theta	(C3)
	Alpha	(T3), (C3), (CZ), (C4), (T4), (T5), (P3), (PZ), (P4), (T6), (O1), and (O2)
	Delta/alpha	(T3), (C3), and (CZ)
	Theta/beta	(C4)
	Theta/delta	(P3, T6)
	Delta	(C3), (CZ), (C4), (T4), (T5), (P3), (PZ), (P4), (T6), (O1), and (O2)
	Theta	(CZ), (P4), and (O2)
	Delta/theta	(T3), (T4), (P3), (PZ), (P4), and (T6)
ΔEEG_A_	Alpha/theta	(T3)
	Theta/alpha	(T3), (C3), (T5), (P3), (PZ), and (P4)
	Alpha/delta	(T6)
	Beta/theta	(C4), and (P4)
	Alpha	(T6)
	Delta/alpha	(P4, T6), (T6, O1), and (T6, O2)
	Theta/beta	(T3, CZ, T4, O1, O2)
ZCI	Alpha/theta	(C3), (P3), (PZ), and (P4)
	Theta/alpha	(C3), (P3), and (P4)
	Alpha/delta	(C3, P4, T6, O1)
	Alpha	(T6, O1), and (T6, O2)
	Delta/alpha	(C3, P3, O1)
Coherence	Alpha/theta	(Fp2-F4, Fz-F8, F4-T4, F8-P4, T3-T4, T3-P4)
	Beta/theta	(Fp2-F4, F4-F8, F4-T3, F4-T4)
	Theta/beta	(F4-F8, F4-T3, F4-T4)

**Table 7 brainsci-11-01026-t007:** Panel of robust EEG biomarkers.

Method	EEG Biomarker Feature	EEG Channel	Sen. %	Spec. %	Category
ApEn	Alpha	P3, T6, O1	100	83	EEG complexity
ApEn	Alpha/theta	C4	100	100	
HFD	Alpha	T5, PZ, T6, O1, O2	88	100	
HFD	Theta/alpha	T3, T5	100	100	
LZC	Alpha	C3, CZ, PZ, T6, O1	88	100	
LZC	Alpha/delta	CZ, C4, T4, P3, P4, T6	100	83	
TsEn	Alpha/delta	C3, T6	100	91	
TsEn	Alpha/theta	T3, C3, CZ, P3, T6, O1	86	90	
ZCI	Alpha	T6, O1, O2	88	100	EEG slowing
ZCI	Alpha/theta	C3, P3, PZ, P4	100	100	
ZCI	Delta/alpha	C3, P3, O1	100	91	
ΔEEG_A_	Alpha/delta	T6	100	100	
ΔEEG_A_	Theta/alpha	T3, C3, T5, P3, PZ, P4	100	100	
ΔPS	Alpha	T3, C3, P3, P4, T6	100	100	
ΔPS	Alpha/delta	T3, T4, P3, P4, O1	100	100	
ΔPS	Alpha/theta	T3, C3, T4, P3, PZ, P4	100	100	
Coh	Alpha/theta	Fp2-F4, Fz-F8, F4-T4, F8-P4, T3-T4, T3-P4	86	90	EEG connectivity
		Combined model	100	100	

**Table 8 brainsci-11-01026-t008:** Changes in EEG signal due to AD for all the 17 robust EEG biomarker panels.

Category	Method	Biomarker Feature	AD	Norm
EEG complexity	ApEn	Alpha		
	ApEn	Alpha/theta		
	HFD	Alpha		
	HFD	Theta/alpha		
	LZC	Alpha		
	LZC	Alpha/delta		
	TsEn	Alpha/delta		
	TsEn	Alpha/theta		
EEG slowing	ΔPS	Alpha		
	ΔPS	Alpha/delta		
	ΔPS	Alpha/theta		
	ZCI	Alpha		
	ZCI	Alpha/theta		
	ZCI	Delta/alpha		
	ΔEEG_A_	Alpha/delta		
	ΔEEG_A_	Theta/alpha		
EEG connectivity	Coh	Alpha/theta		

**Table 9 brainsci-11-01026-t009:** Number and distribution of panels with one, two, three, and four biomarkers.

No. of Biomarkers in a Panel	No. of Panels
1	69
2	2346
3	52,394
4	864,501
Total	919,310

**Table 10 brainsci-11-01026-t010:** The performance of the alpha band for the TsEn method.

EEG Channel	Sen. %	Spec. %	Acc. %	F-Measure %	MCC	PPV %	NPV %
C3	50.00	66.67	58.82	53.33	0.17	57.14	60.00
P3	35.71	33.33	35.29	47.62	−0.24	71.43	10.00
C3, P3	83.33	81.82	82.35	76.92	0.63	71.43	90.00
O2	30.77	25.00	29.41	40.00	−0.38	57.14	10.00
C3, O2	55.56	75.00	64.71	62.50	0.31	71.43	60.00

**Table 11 brainsci-11-01026-t011:** Performance of biomarker panels consisting of one, two, three, four, five, and six biomarkers for dataset B.

EEG Biomarkers	Sen. %	Spec. %
[ΔEEG_A_(alpha/delta(T6))]	100	60
[ΔEEG_A_(theta/alpha(T5))]	61.71	100
[ΔEEG_A_(alpha/delta(T6)), ΔEEG_A_(theta/alpha(T5))]	100	61.54
[ZCI(alpha/theta(P3)), ΔEEG_A_(theta/alpha(T3))]	54.84	100
[ZCI(alpha/theta(P3)), ZCI(delta/alpha(P3)), ΔEEG_A_(theta/alpha(T3))]	54.84	100
[ZCI(alpha/theta(P3)), ΔEEG_A_(theta/alpha(T3)), ΔEEG_A_(theta/alpha(T5))	54.84	100
[TsEn(alpha/theta(T6)), ZCI(delta/alpha(P3)), ΔEEG_A_(alpha/delta(T6)), ΔEEG_A_(theta/alpha(T5))]	100	68.57
[ZCI(alpha/theta(P3)), ZCI(delta/alpha(P3)), ΔEEG_A_(theta/alpha(T3)), ΔEEG_A_(theta/alpha(T5))]	54.84	100
[TsEn(alpha/theta(T6)), ZCI(alpha/theta(P3)), ZCI(delta/alpha(P3)), ΔEEG_A_(theta/alpha(T3)), ΔEEG_A_(theta/alpha(T5))]	54.84	100
[ZCI(alpha/theta(P3)), ZCI(delta/alpha(P3)), ΔEEG_A_(alpha/delta(T6)), ΔEEG_A_(theta/alpha(T3)), ΔEEG_A_(theta/alpha(T5))]	80.95	100
[TsEn(alpha/theta(T6)), ZCI(alpha/theta(P3)), ZCI(delta/alpha(P3)), ΔEEG_A_(alpha/delta(T6)), ΔEEG_A_(theta/alpha(T3)), and EEG_A_(theta/alpha(T5))]	85	100

## Data Availability

Due to privacy and ethical concerns, neither the data nor the source of the data can be made available.

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
