# Peer review of "Robust EEG Based Biomarkers to Detect Alzheimer’s Disease"

_brainsci, 2021, doi:10.3390/brainsci11081026_

Round 1

Reviewer 1 Report

This study uses machine learning on 4 data sets containing AD patients and age-matched controls in order to identify combined biomarkers for AD. The data sets are well-defined, and the complex analyses seem to be run in a reasonable manner that should yield statistically valid results. These results are potentially useful to researchers and clinicians, although I do have concerns about the extension of these results for clinical use due to the computational complexity. Certainly the steps could be patented and simplified as a commercial application. My concerns are listed below, and primarily deal with the introduction that needs to be expanded to include hypotheses that are clearly connected to the extant research literature, and the Discussion, that needs to summarize better how this research adds to existing knowledge.

Ln. 53: and represents 70-80% of 53 all dementias [18]. Redundant with ln. 39.

Ln 141: "No such comprehensive EEG biomarker study for AD has hitherto been carried out." I would direct the authors to the paper below. It is in the references section, and does represent a multiple biomarker EEG study for AD. I think the claim is too strong.

"Poil, S.-S.; De Haan, W.; van der Flier, W.M.; Mansvelder, H.D.; Scheltens, P.; Linkenkaer-Hansen, K. Integrative EEG Biomarkers Predict Progression to Alzheimer’s Disease at the MCI Stage. Front. Aging Neurosci. 2013, 5, doi:10.3389/fnagi.2013.00058."    The Rossini and Babiloni papers are also in this same family of papers.   What the reader needs in the introduction is a review of what work has been done using multiple biomarkers, and specifically what this paper adds that is novel to the field. Is it simply the choice of the exact biomarkers, or the sheer number, or the addition of a novel biomarker that has never been tried?   Major concern: There are no hypotheses laid out at the end of the Introduction. These must be included. This is not an engineering journal, but a biosciences journal. As such, even though this is a data driven paper, there needs to be a discussion of the hypotheses that limits the scope of the analyses. This has kind of a "kitchen sink" feeling, where there is just every type of possible biomarker thrown in with no discussion of their overlap.   I would like to authors to speak in the introduction, a priori about the possible overlap between biomakers. How much should each one add to the predictive accuracy? The question is of unique versus shared variance really. So I am not looking for a lengthy dissertation on this, just a reasonable review that tells the reader WHY some of these were selected and not others, and also how they different measures are complimentary and not redundant.

 You do mention:

1) slowing of the EEG 

2) reduction in EEG complexity and

3) reduction in EEG coherence, but not WHY these are unique or important.

Ln 252: Define SVM for the reader on its first occurrence.

Please provide parameters and assumptions if your SVM initially in the methods.

Please define LDA for the reader in its first occurrence.

Please change "Discussions" to Discussion.

Please move: 512 to 533 to the discussion section, as it is data interpretation.

In section 3.5, can you please remind the reader exactly how biomarkers were combined? Is this multiple logistic regression, or some other approach?

Ln 754-777. Please do not simply list studies and their findings. The Discussion is supposed to be just that, a discussion, in which a logical argument is constructed that leads the reader to a conclusion. Please include connecting and transition sentences that help make the importance of each finding clear.

Why would temporal and parietal lobe channels do the best? Please connect this to literature or attempt an explanation. Could it simply be proximity to cerebral ventricles, or some cranial features, or are they position to pick up on dysfunctional neural systems?

Reviewer 2 Report

Summary of the manuscript

Al-Nuaimi et al carefully characterizes EEG biomarkers of Alzheimer’s disease (AD). Specifically, the authors examine a large set of possible EEG-related components to identify a panel of six biomarkers that can detect AD with high accuracy.

The strengths of this manuscript are in two-fold. First, the analytical framework developed in the manuscript is clearly explained. I especially appreciate the reference to all relevant literature accompanying each component of the framework. The clarity with which the authors describe the analyses makes it extremely easy to follow the method setup as well as understand the results. Second, the topic discussed is well-motivated and has important implications for clinical practice as well as basic science research investigating the neural bases of cognitive functions in health and disease.

Having said this, I found certain information regarding the link between the investigated biomarkers and their biological relevance to cognition to be missing. I believe that the manuscript could benefit from more explicit discussion of how the identified biomarkers could reflect specific aspects of disrupted cognitive processing in AD.

Comments:

  1. How do you think the number of EEG electrodes contribute to the reported findings? In other words, how do you think the proposed biomarkers would do in the case where EEG from AD patients is recorded using larger number of electrodes (e.g., 72 channels)? It is possible that greater density in recording channels reduce the classification accuracy in certain markers (e.g., coherence-related measures) while increase accuracy in other markers (e.g., frequency-specific measures)?
  2. I appreciate the thorough explanation of how each EEG biomarker has previously been evaluated in the context of other neurological conditions such as ADHD. However, the manuscript lacks discussion of what neural aspects of neural dynamics/processes these biomarkers may capture. Are the differences region-specific? For example, why one may expect that AD-dependent changes in the frequency-band specific manner. In fact, several studies already cited in the manuscript in the context of ADHD have explicitly investigated occipital alpha as a maker for attention-related processing. Another frequency band activity highly relevant to information processing and memory-related functions is theta. Below are some example literatures showing the importance of frontal theta in such processes:
  • Summerfield et al., Frontiers in human neuroscience, 2011. Human scalp electroencephalography reveals the repetition suppression varies with expectation
  • Rungratsameetaweemana et al., Journal of Neuroscience, 2018. Expectations do not alter early sensory processing during perceptual decision-making.
  • Ergen et al., International Journal of Psychophysiology, 2014. Time-frequency analysis of the event-related potentials associated with the Stoop task.

Here are some example findings where beta-band activity is linked to motor-related cognition:

  • Wyart et al., Neuron, 2012. Rhythmic fluctuations in evidence accumulation during decision making in the human brain.
  • Donner et al., Current Biology, 2009. Buildup of choice-predictive activity in human cortex during perceptual decision making.

Finally, what are the possible neural underpinnings of frequency ratio? Even if certain biomarkers may not have been explicitly linked to neural functions, explicit speculation/proposals on how the identified biomarkers could help elucidate what neural dynamics are affected in AD could be extremely helpful.

  1. The topic investigated in the manuscript is important and timely. Although the illustrated analytical framework is developed for AD, it can likely be generalized to examine the neural dynamics associated with other neurological conditions such as epilepsy and Schizophrenia. Incorporation of such formulation will benefit future studies investigating AD and other neurological conditions from neurobiological as well as computational perspectives.
